# Distribution of Antibiotic Resistance Genes in *Kocuria* Species

**DOI:** 10.3390/antibiotics14101041

**Published:** 2025-10-17

**Authors:** Elizaveta M. Pleshko, Marina V. Zhurina

**Affiliations:** Laboratory of Viability of Microorganisms, Research Center of Biotechnology RAS, Moscow 119071, Russia; lizapleshko@yandex.ru

**Keywords:** biofilm, *Kocuria*, antibiotic resistance

## Abstract

Backgroud. *Kocuria* are widespread Gram-positive bacteria. Although they are traditionally classified as non-pathogenic, recent studies have shown that they can cause problems in various fields, from livestock and aquaculture to medicine. This has led to an increased need to understand their antibiotic resistance mechanisms in order to combat them. Methods. To study the determinants of *Kocuria* antibiotic resistance, we used bioinformatics methods. To identify antibiotic resistance genes, we retrieved the complete genome sequences of *Kocuria* strains from the RefSeq database and screened them for antibiotic resistance determinants with different mechanisms of action. We also studied *Kocuria* strains in more detail: we sequenced whole genomes of *K. carniphila* 988, *K. rhizophila* 155, *K. rosea* 394 and *K. rosea* 397, and, in addition to bioinformatics studies, and tested five strains for their ability to grow in the presence of antibiotics. Results. For these five strains, the presence of antibiotic resistance genes in their genomes correlated well with the observed resistance to the corresponding antibiotics: all 5 studied strains have a high level of resistance to chloramphenicol, in addition, *K. carniphila* 988 is highly resistant to azithromycin and avilamycin. Conclusions. Therefore, it has been demonstrated that antibiotic resistance genes are present in many *Kocuria* genomes and these genes are functional in the strains we have studied.

## 1. Introduction

*Kocuria* spp. are Gram-positive bacteria from the family *Micrococcaceae* and the phylum *Actinobacteria*. Although they are not considered pathogenic, there has been a growing number of reports of concomitant diseases caused by these bacteria [1]. *Kocuria* spp. are normal representatives of the microbiota of the skin and mucous membranes in humans and animals. Therefore, most clinical microbiology laboratories have until recently ignored such bacteria as sample contaminants. With the increasing reports of infections associated with these bacteria, it is now important for clinical microbiologists to identify these microorganisms and consider their susceptibility/resistance to antibiotics [2,3].

*Kocuria* spp. can cause a variety of serious diseases, including bacteremia, endocarditis, endophthalmitis, peritonitis, keratitis, infections of the respiratory tract, nervous system, bones, joints, and urinary tract [4].

Numerous strains of the genus *Kocuria* have been shown to be resistant to antibiotics. According to a narrative review based on 73 studies (including 102 patients with *Kocuria* spp. infections), resistance to macrolides was observed in 52.4% of cases, penicillin resistance in 51%, clindamycin resistance in 44.7%, aminoglycoside resistance in 36.2%, trimethoprim–sulfamethoxazole resistance in 30.4%, quinolone resistance in 28.4%, cephalosporin resistance in 17.9%, rifampicin resistance in 16.7%, vancomycin resistance in 7%, and tetracycline resistance in 6.7% [4]. This can be a serious problem in the treatment of infections caused by *Kocuria* spp.

Additionally, the genus *Kocuria* is known for its ability to form persistent forms that can survive in various adverse conditions. These forms can disseminate within the environment and persist despite antibiotic treatment in the host organism [5,6]. Therefore, the potential for infections caused by *Kocuria* species should not be underestimated.

*Kocuria* species are widely distributed and occupy diverse ecological niches. In particular, they are often found in habitats where they are likely to come into contact with antibiotics, such as meat processing plants and household waste. This may be the reason for the spread of antibiotic resistance among *Kocuria* spp. [7,8]. Furthermore the existence of *Kocuria* carrying antibiotic resistance genes has been shown in sewers [9], in fish cage aquaculture [10] and in raw milk of cows [11].

*Kocuria* spp. are often mentioned among microorganisms isolated from meat processing plants, as well as from carcasses and meat products [7,12]. Microorganisms belonging to the normal biota of farm animals or residing on objects associated with the food industry may develop antibiotic resistance. This can lead to the transmission of pathogenic antibiotic-resistant bacteria from animals to humans [13].

Another significant ecological niche that facilitates the dissemination of antibiotic resistance genes is municipal solid waste, where antibiotics end up as a result of improper disposal of medicines. Municipal waste provides an opportune environment for the proliferation of microorganisms, including potential pathogenic species. Another significant ecological niche that facilitates the dissemination of antibiotic resistance genes is municipal solid waste, where antibiotics end up as a result of improper disposal of medicines. Municipal waste provides an opportune environment for the proliferation of microorganisms, including potential pathogenic species. Horizontal gene transfer of antibiotic resistance from resistant to susceptible microorganisms is highly probable under these conditions. Subsequently, resistant microorganisms can readily enter the environment. Municipal waste serves as a reservoir of particulate matter that can be transported over long distances. Areas where municipal waste is temporarily stored are typically located in close proximity to human settlements, including regions with high population densities. Consequently, there is a risk of inhalation of particulate matter containing antibiotic-resistant pathogens. Recent data indicates that members of the genus *Kocuria* are among the most prevalent antibiotic-resistant bacteria found in particulate matter derived from municipal waste [8].

Thus, although cases of infections caused by *Kocuria* spp. are rare and relate mainly to people with weakened immune systems, the problem of antibiotic resistance of these bacteria cannot be underestimated. An explanation of the mechanisms underlying resistance to various antibiotics may be useful to address this issue. In this study, we aim to learn more about the antibiotic resistance of bacteria *Kocuria* spp. The strains were chosen for testing because they originated from different ecotopes: we isolated strains *K. rhizophila* 155 and *K. carniphila* 988 from meat processing plants that regularly conduct antibacterial treatments. The other three strains were isolated from soil and the rhizosphere, meaning they could have come into contact with fungal antibacterial compounds, but not with human-synthesized preparations. To achieve our aim, we are, on the one hand, determining the complete genomic sequences of four previously unsequenced bacteria from this genus (*K. rhizophila*, *K. carniphila* and two different strains of *K. rosea*). On the other hand, we investigated the phenotypic expression of antibiotic resistance by culturing five strains of *Kocuria* in the presence of antibiotics with different mechanisms of action. So, we analyzed the distribution of antibiotic resistance genes within the genus *Kocuria* and compared the presence of antibiotic resistance genes with the phenotypic antibiotic resistance for several test strains. The acquired data has been systematically compiled and is intended to be of value to a broad spectrum of researchers, encompassing both specialists in bacteriology and those engaged in the study of antibiotic resistance phenomena.

## 2. Results

### 2.1. Genomes Quality Assessment

We assessed genome completeness and contamination of four *Kocuria* genomes. All the genomes were of excellent quality (>95% completeness and <5% contamination). For all genomes the dDDH value was more than 70%, confirming the taxonomic placement of studied strains. These results are provided in Table 1.

### 2.2. Antibiotic Resistance Genes Identification

Complete genomes with protein annotations for 258 *Kocuria* strains were downloaded from the RefSeq database. We found genes involved in resistance to macrolides, aminoglycosides, beta-lactam antibiotics, rifampicin, tetracycline, vancomycin, quinolones and orthozomycins. Only proteins that modify antibiotics or their targets were considered, since these determinants provide the highest resistance and can effectively spread through horizontal gene transfer. The search results are given in Appendix A.

The following antibiotic resistance genes were found: macrolides-23S rRNA (guanosine(2251)-2′-O)-methyltransferase (RGMT) and macrolide-2′-phosphotransferase (MPT); chloramphenicol-23S rRNA (adenine(2503)-C(2))-methyltransferase RlmN, β-lactam antibiotics-β-lactamase, aminoglycosides-aminoglycoside-3′-phosphotransferase (APH), rifampicin-NAD(+)-rifampin-ADP-ribosyltransferase and rifampin monooxygenase; avilamycin-23S rRNA (uridine(2479)-2′-O)-methyltransferase (RUMT). Genes that may be involved in resistance to cephalosporins, quinolones, vancomycin, and tetracycline were not found in any of the protein annotations. The number of strains whose genomes contain at least one copy of each gene is shown in Figure 1.

### 2.3. Phylogenetic Tree Construction

The distribution of resistance genes among *Kocuria* spp. was compared with the phylogeny of the genus. For this purpose, the *Kocuria* phylogenetic tree was constructed (see Appendix A). For tree construction we used 181 genomes of *Kocuria* strains for which the species was identified.

Almost all *Kocuria* species form monophyletic groups on the resulting tree. The branches of the tree corresponding to the species have great bootstrap support. However, *Kocuria turfanensis* and *Kocuria oceani* are very close to each other. Representatives of these species are not divided into two groups, so perhaps they should be classified as one species. In addition, the strains of *Kocuria artinae* form a paraphyletic group. They are very close to *Kocuria carniphila* and should probably be assigned to this species. There are also individual strains that appear to have been misclassified. *Kocuria varians* 80 should belong to the species *K. salsicia*, *Kocuria rhizophila* TNDT1–to the species *K. flava*, *Kocuria rosea* S-A3–to *K. dechangensis*. However, the reclassification of strains is not the purpose of this study, and building a phylogenetic tree may not be sufficient to change the systematic position of the strains.

For further work, we constructed a phylogenetic tree based on reference genomes of *Kocuria* spp. The resulting tree matches the configuration of the tree built across all genomes. In Figure 2 the phylogenetic tree of *Kocuria* and antibiotic resistance genes that are present in genomes of strains belonging to each species.

The majority of *Kocuria* strains have genes involved in resistance to macrolides. Many strains have both RGMT and MPT. A number of strains from the group formed by the species *K. gwangalliensis*, *K. artinae*, and *K. carniphila* have two macrolide-2′-phosphotransferases. The RlmN gene, which provides resistance to chloramphenicol, is present in almost all strains. Apparently, *Kocuria* inherited it from their common ancestor. Genes involved in resistance to other antibiotics are rarer. Representatives of the *K. massiliensis* species and the *Kocuria rhizophila* RF strain have B-lactamase. The APH gene, which provides resistance to aminoglycosides, is present in individual representatives of the species *K. coralli*, *K. rosea*, *K. carniphila* and *K. rhizophila*. Some strains of *K. rosea* and *K. carniphila* have two APH genes, and the *Kocuria rhizophila* UMB0131 strain has three of these genes. Determinants of rifampicin resistance are also rare among *Kocuria* spp. Representatives of the species *K. gwangalliensis* and *K. carniphila* have RRT. RM is present only in the strains *Kocuria dechangensis* CGMCC 1.12187 and *Kocuria rosea* TA 28, as well as in several strains unclassified to species. Only a single strain of *Kocuria carniphila* 988 has a determinant of resistance to avilamycin RUMT.

We previously showed the presence of the RUMT gene in genome *K. carniphila* 988 by using the RAST server for genome annotation [14]. In the annotation automatically obtained using the PGAP program, this protein is identified as TrmH family RNA methyltransferase (WP_369067065.1). Only *K. carniphila* 988 has this gene. The BLAST search (BLAST 2.15.0) for this protein indicates that it is most closely related to genes from other members of the *Micrococcaceae* family. Apparently, the protein gene was recently acquired by strain *K. carniphila* 988 during horizontal transfer. Avilamycin is an antibiotic widely used in the food industry for the treatment of farm animals. The *K. carniphila* 988 strain was isolated at a meat processing plant. Apparently, it was regularly exposed to avilamycin during its evolution when rearing animals before entering meat processing, which is why the presence of the RUMT gene provided it with an adaptive advantage.

### 2.4. Antibiotic Resistance Assessment

In order to check whether the presence of determinants of antibiotic resistance is manifested in the phenotype, minimum concentrations inhibiting the growth of the studied strains by at least 50% (MIC50) were determined. They are shown in Table 2.

*K. rhizophila* ATCC 9341 and *K. rhizophila* 155 strains are very sensitive to azithromycin. Their growth is suppressed at the lowest concentrations of the antibiotic. The *K. carniphila* 988 has shown high levels of azithromycin resistance. The growth of *K. carniphila* 988 is not suppressed even at a concentration of 1024 µg/mL. *K. rosea* 394 and *K. rosea* 397 strains have medium resistance. *K. carniphila* strain 988 has the largest number of macrolide resistance genes: one GMT gene and two MPT genes. Other studied strains also have macrolide resistance genes, but they are probably less active. Apparently, during the evolution of the *Kocuria* species *K. carniphila* and the species closest to it, *K. artinae* and *K. gwangalliensis*, which also have two macrolide-2′-phosphotransferase genes, were most affected by macrolides.

All strains have shown high levels of chloramphenicol resistance which corresponds to the presence of RlmN. All strains have an average resistance to kanamycin. The most resistant strain is *K. rhizophila* 155. Its growth is suppressed at a concentration of 64 µg/mL. However, it does not have the APH gene in its genome, so kanamycin resistance is provided by some other mechanisms.

All strains are very sensitive to avilamycin, except *K. carniphila* 988, which has shown high resistance. Its growth is not suppressed even at a concentration of 1024 µg/mL, while the growth of other strains is suppressed already at a concentration of 1 µg/mL. This is consistent with the fact that *K. carniphila* 988 is the only *Kocuria* strain with the RUMT gene.

All the studied strains are sensitive to other antibiotics. In most cases, this is consistent with the absence of determinants of antibiotic resistance in their genomes. However, the genome of *K. carniphila* 988 contains the RRT gene, and the genomes of *K. rosea* 397 and *K. rosea* 394 contain the RM gene, the presence of which should ensure resistance to rifampicin. But despite this, both strains are sensitive to this antibiotic. In addition, *K. carniphila* 988 is sensitive to kanamycin despite the presence of the APH gene in its genome. Apparently, these genes are inactive.

## 3. Discussion

We searched for antibiotic resistance genes from several most commonly used groups and investigated the manifestation of these genes in the phenotype by conducting antibiotic sensitivity tests. Based on our results, we can conclude that resistance to macrolides and chloramphenicol is widespread among *Kocuria* spp. Strains resistant to beta-lactam antibiotics, aminoglycosides, and rifampicin should be much less common, which is consistent with the data of other researchers [4]. Resistance to avilamycin is a unique feature of the *K. carniphila* 988. Resistance to tetracycline, vancomycin and quinolones is unusual for *Kocuria* spp.

In most cases, the presence of antibiotic resistance in the studied strains is consistent with the presence of resistance genes in their genomes. However, differences in resistance to aminoglycosides and macrolides in different strains cannot be explained solely by the presence or absence of certain genes in their genomes. The available genes may differ in their level of activity. In addition, more complex mechanisms may be involved, such as differences in cell permeability to antibiotics, the presence of drug efflux pumps, and differences in metabolic pathways.

The results obtained are generally consistent with the data on the antibiotic resistance of clinical *Kocuria* spp. isolates [4].

However, resistance to certain antibiotics, such as beta-lactam and rifampicin, is more common than would be expected based solely on the spread of determinants of antibiotic resistance among *Kocuria* strains. In addition, there are known cases of resistance to quinolones, vancomycin and tetracycline, which cannot be explained by the presence of certain resistance determinants in the genomes. However, it should be borne in mind that in most cases, infections were caused by representatives of the *K. kristina* (46.1%). This species has recently been reclassified into *Rothia kristina* [15]. Therefore, in our work we did not analyze the genomes of strains belonging to this species. Recently, a group of researchers proposed another reclassification of the *Micrococcaceae* family, which may lead to changes in the future [16].

In addition, discrepancies in the observed antibiotic resistance and the presence of relevant determinants in the genomes may be explained by inaccuracies in protein annotations. To determine the functions of many genes, it is not enough to analyze the functions of their homologues from other organisms. Many genes are classified as genes of hypothetical proteins, while for others it is possible to identify only the protein families to which their products belong. Among these genes, there may be previously unknown determinants of antibiotic resistance. In some cases, they can be identified by using different annotation programs, as the results obtained by different programs vary slightly. Some genes are annotated with varying accuracy. Thus, the RUMT gene in the *K. carniphila* 988 genome was identified by the RAST server as 23S rRNA (uridine(2479)-2′-O)-methyltransferase, and by the PGAP program as TrmH family RNA methyltransferase. In our work, annotations obtained using PGAP and attached to complete genomes were used to search for determinants of antibiotic resistance, since they are created automatically and are available for the largest number of strains.

Our work provides a general idea of the distribution of antibiotic resistance genes among *Kocuria* spp. Strains with resistance genes in their genomes do not form monophyletic groups, which suggests that these genes have been independently acquired by members of the genus as a result of horizontal gene transfer. In some cases, this seems to have happened quite recently, since in many cases, single representatives of species have resistance genes. Thus, the RUMT gene was most likely acquired by *K. carniphila* 988, as the presence of resistance to avilamycin provides an adaptive advantage in ecological niches related to the food industry.

From the data obtained, it can be concluded that the horizontal transfer of genes associated with antibiotic resistance is quite active between *Kocuria* spp. and other microorganisms. Since *Kocuria* are widespread and often found in habitats where they are highly likely to come into contact with human pathogens, knowledge about the determinants of antibiotic resistance present in these microorganisms is of great practical importance.

The importance of the presence and prevalence of antimicrobial resistance genes in strains of the genus *Kocuria* is due to the fact that the existence of these bacteria in multispecies biofilms has been shown (at least in food production [17,18]). Such communities have a resistome [19]. The antibiotic resistome is the collection of all the antibiotic resistance genes, including those usually associated with pathogenic bacteria isolated in the clinics, non-pathogenic antibiotic producing bacteria and all other resistance genes. In consequence, each microorganism contributes to the resistance of this multispecies community. Thus, knowledge of the determinants of antibiotic resistance in *Kocuria* spp. is of great practical importance for the food industry and public health.

## 4. Materials and Methods

### 4.1. Bacterial Strains and Culture Conditions

The cultural work was carried out with five *Kocuria* strains: *K. rhizophila* ATCC 9341 from the American Type Culture Collection, *K. rosea* IEGM 397 and *K. rosea* IEGM 394 from the “IEGM Regional Specialized Collection of Alkanotrophic Microorganisms”, *K. rhizophila* 155 and *K. carniphila* 988, isolated from a meat processing plant and deposited into “Collection of unique and extremophilic microorganisms of various physiological groups for biotechnological purposes UNIQEM”. Cultures of *K. rhizophila* ATCC 9341, *K. rhizophila* 155 and *K. carniphila* 988 were grown at 30 °C and stirring at 150 rpm, cultures of *K. rosea* 394 and *K. rosea* 397 were grown at room temperature and stirring at 250 rpm. LB medium (Lennox, Dia-M, Moscow, Russia) was used for the cultivation of microorganisms.

All the strains were stored at room temperature in tubes with semiliquid lysogeny broth supplemented with 0.4% agar covered with sterile mineral oil. The purity of the cultures was checked by streaking to individual colonies on a Petri dish with agar LB medium (2% agar-agar, Dia-M, Moscow, Russia)).

To prepare inoculum, a single colony from the Petri dish was inoculated into a 50 mL Erlenmeyer flask with 20 mL of sterile LB and incubated to obtain an optical density (OD) at 540 nm of 2.0.

A unique aspect of growing *Kocuria* cultures [6,20] is the formation of cell aggregates, and sensitivity to antibiotic depends on the size of the aggregates. Thus, cells within these aggregates are partially protected from antibiotic exposure Attempts to dissociate the aggregates into individual cells result in a cessation of culture growth. Necessitating a high inoculum containing aggregates for the cultivation of both studied *K. rosea* strains. Moreover, efforts to achieve complete inhibition of growth yield highly variable results. To address this issue, a modified methodology was implemented. Following cultivation, the cell aggregates were carefully dispersed before measuring optical density. After that, determining the antibiotic concentration required to inhibit growth by 50% (MIC50). This approach was employed to compare the antibiotic sensitivity of the strains among themselves.

### 4.2. Genomes and Bioinformatics Analysis

To search for antibiotic resistance genes, all complete genomes of 258 *Kocuria* strains along with the corresponding protein annotations, were downloaded from the RefSeq database [21].

The genomes of *K. rhizophila* 155 and *K. carniphila* 988 strains were previously obtained by us [14] and deposited in the NCBI Assembly database with the identifiers K. *rhizophila* 155-ASM4102137v1 and *K. carniphila* 988-ASM4102122v1. These genomes were annotated using the RAST server [22].

We obtained full genomes of two *Kocuria rosea* strains from the IEGM Regional Specialised Collection of Alkanotrophic Microorganisms: *K. rosea* 394 and *K. rosea* 397. We extracted DNA using Magen HiPure Bacterial DNA Kit according to the manufacturer’s instructions (briefly, it is a silica column-based method involves four main steps: cell lysis, DNA binding to the column, washing to remove contaminants, and elution of the purified DNA), and sequenced genomes with the support of the NextGenSeq. Obtained genomes were assembled as described previously [14] and deposited into the NCBI Assembly database with following identifiers: *K. rosea* 394-ASM5104275v1 and *K. rosea* 397-ASM5058636v1.

Genome completeness and contamination was assessed using MIGA [23,24]. To confirm the taxonomic placement, we assessed dDDH values against species type strains using TYGS [25].

The VBCG program was used to construct the phylogenetic tree of the genus *Kocuria* [26]. The MEGA11 [27] program was used for visualization of phylogenetic trees.

### 4.3. Search for Antibiotic Resistance Genes

To search for antibiotic resistance genes, we downloaded protein sequences from the Uniprot database [28], which, according to literature data, are associated with bacterial antibiotic resistance. The BLAST+ program [29] and Python scripts were used to search for antibiotic resistance proteins in protein annotations of *Kocuria* spp. genomes. For each gene we obtained the information, in which genomes it is present. Next, we compared the presence of these genes in genomes with the phylogeny of *Kocuria*. Using Python scripts (Python 3.14.0), the presence of antibiotic resistance determinants was compared with the phylogeny of the genus *Kocuria.* Phylogenetic trees of the found proteins were constructed in the MEGA11 program [27].

In order to check whether strains that have antibiotic resistance genes in their genomes are resistant to relevant antibiotics, antibiotic sensitivity tests were performed on five *Kocuria* strains.

### 4.4. Antibiotics

To test for antibiotics, an inoculum grown in rocking flasks was seeded into wells of 96-well flat-bottomed plates (Kirgen, Haikou, China) containing 200 µL of nutrient medium with various concentrations of antibiotics. The following antibiotics were used for the tests: azithromycin (JSC VERTEX, Saint-Petersburg, Russia), chloramphenicol (Bioinnlabs, Rostov-on-Don, Russia), ampicillin (RUE BELMEDPREPARATY, Minsk, Belarus), kanamycin (JSC BIOCHEMIK, Moscow Region, Russia), rifampicin (Kraspharma, Krasnoyarsk, Russia), avilamycin (GlpBio Technology, Montclair, CA, USA), tetracycline (JSC BIOCHEMIST, Moscow Region, Russia), ciprofloxacin (Sintez Pharmaceuticals, Irkutsk, Russia), vancomycin (RUE BELMEDPREPARATY, Minsk, Belarus). The stock solutions of antibiotics were sterilized by filtration through a 0.22 µm pore-size filter and stored frozen (−20 °C) for 3 months. Each well was inoculated with 5 µL for of *K. rhizophila* ATCC 9341, *K. rhizophila* 155 or *K. carniphila* 988 and 20 µL for *K. rosea* 394 or *K. rosea* 397. The plates were cultured with shaking for 3 days at 30 °C. To assess the growth of crops, the optical density in the wells of the tablet was measured at a wavelength of 540 nm. Based on the data obtained, the minimum concentrations inhibiting the growth of the studied strains by at least 50% (MIC50) were determined.

### 4.5. Statistical Analysis

The nonparametric Mann–Whitney U-test was used to assess the significance of differences (they were considered significant at *p* < 0.05). All measurements were performed in at least three statistical replications. The software of Microsoft Excel was used for processing of the data.

## 5. Conclusions

We showed that *Kocuria* spp. have genes that should be involved in resistance to a number of antibiotics. We showed that the presence of these antibiotic resistance genes in their genomes consistent with observed resistance to corresponding antibiotics for five studied strains of *Kocuria*. We also found that genes conferring antibiotic resistance are present in many *Kocuria* genomes, likely acquired through horizontal gene transfer. The most common resistance genes are those associated with macrolides and chloramphenicol, while some strains also harbor genes linked to resistance to aminoglycosides, beta-lactams, and rifampicin. However, no genes related to resistance to tetracycline, quinolone, or vancomycin have been detected in *Kocuria* yet. These findings may be useful for fundamental research into antibiotic resistance as well as in situations where *Kocuria* strains cause problems and need to be controlled in industry and medicine.

## Figures and Tables

**Figure 1 antibiotics-14-01041-f001:**
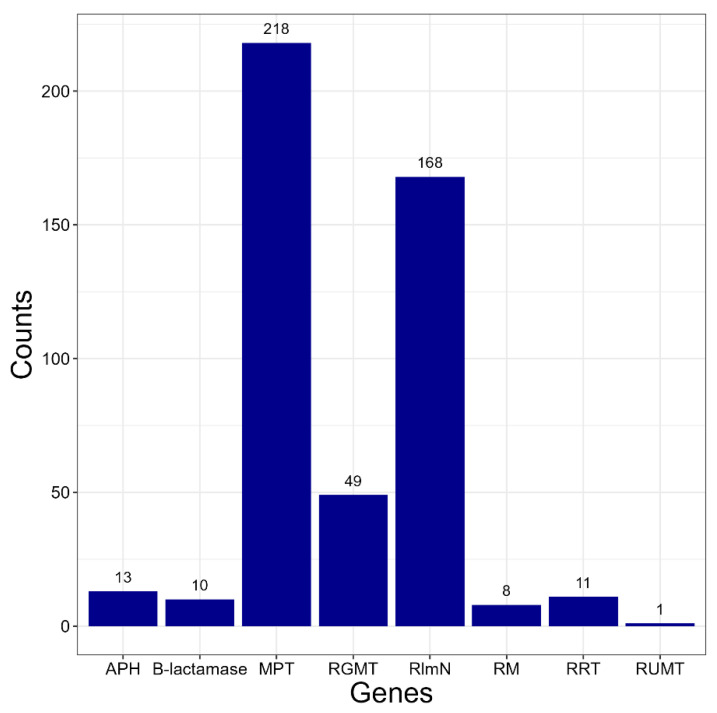
The number of *Kocuria* strains whose genomes contain at least one copy of each gene. Genes: APH aminoglycoside-3′-phosphotransferase; MPT-macrolide-2′-phosphotransferase; RGMT-23S rRNA (guanosine(2251)-2′-O)-methyltransferase; RlmN-23S rRNA (adenine(2503)-C(2))-methyltransferase; RM-rifampin monooxygenase RRT-NAD(+)-rifampin-ADP-ribosyltransferase; RUMT-23S rRNA (uridine(2479)-2′-O)-methyltransferase.

**Figure 2 antibiotics-14-01041-f002:**
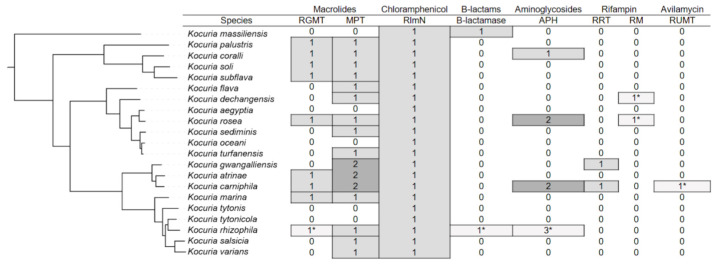
Phylogenetic tree of the genus *Kocuria* with indication of the number of resistance genes in annotations of different species. The numbers marked with an asterisk mean that only one strain belonging to this species has the corresponding determinant. In other cases, many strains have the appropriate genes.

**Table 1 antibiotics-14-01041-t001:** The quality of obtained *Kocuria* genomes.

DDDH d4 Value	Genome Contamination	Genome Completeness	Strain
81.90%	2.80%	98.10%	*K. carniphila* 988
87.90%	1.90%	97.20%	*K. rhizophila* 155
99.90%	0.90%	98.10%	*K. rosea* 394
87.00%	1.90%	98.10%	*K. rosea* 397

**Table 2 antibiotics-14-01041-t002:** Concentrations of antibiotics suppressing the growth of the studied strains by at least 50%, µg/mL.

Antibiotic	Class of Antibiotic	Mechanism of Action	*K. rhizophila* ATCC 9341	*K. rhizophila* 155	*K. carniphila* 988	*K. rosea* 394	*K. rosea* 397
Azithromycin	Macrolides	Binds to the 50S subunit of the ribosome, inhibits translation	1	1	>1024	16	256
Chloramphenicol	Amphenicol	Binds to the 50S subunit of the ribosome, inhibits translation	>1024	1024	>1024	512	1024
Ampicillin	β-lactams	Disrupts cell wall synthesis	1	1	1	1	1
Kanamycin	Aminoglycosides	Bind to the 30S subunit of the ribosome, preventing the peptide elongation	4	64	4	4	4
Rifampicin	Ansamycins	Binds to the β-subunit of DNA-dependent RNA polymerase, preventing its attachment to DNA	1	1	1	1	1
Avilamycin	Orthozomycins	Binds to the 50S subunit of the ribosome, inhibits translation	1	1	>1024	1	1
Tetracycline	Tetracyclines	Binds to 16S rRNA, preventing aminoacyl-tRNA from entering the A-site of the ribosome	1	1	1	1	1
Ciprofloxacin	Quinolones	Type II and IV topoisomerase inhibitor	1	1	1	1	1
Vancomycin	Peptides	Disrupts cell wall synthesis	1	1	1	1	1

## Data Availability

The original contributions presented in this study are included in the article. Further inquiries can be directed to the corresponding authors.

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
