# Peer review of "Distribution of Antibiotic Resistance Genes in Kocuria Species"

_antibiotics, 2025, doi:10.3390/antibiotics14101041_

Round 1
Reviewer 1 Report
Comments and Suggestions for Authors
Dear authors, I consider the subject of the article to be interesting, but it has many deficiencies that prevent me from accepting the article in its current version. Below I share several observations that are required to improve its quality and for its subsequent reconsideration for publication in the journal:
- The title of the article is very conflicting for me, as the search for antibiotic resistance genes is not being conducted through laboratory experiments, so I disagree with its appropriate title. I would also suggest "in Kocuria species," not "among Kocuria spp". In addition, it should be noted whether it was performed on ATCC strains, on meat samples, or on isolates from animals or humans.
- Please better explain information related to the authors' affiliations, including the laboratory and/or department of the Research Center they are affiliated with. In addition, please provide the city and country where the Center is located.
- I suggest rewriting the "Abstract" section to include background, methods, results, and conclusions. This will help readers properly understand what was done in the article.
- Section "1. Introduction" should be improved. I recommend including more characteristics of Kocuria, the antimicrobial resistance mechanisms exhibited by this bacterial genus, and its impact on animal, human, and public health. The terms "flora" or "microflora" are obsolete; they should be replaced with "biota" or "microbiota".
- Section "2. Results" should follow section "1. Introduction" immediately; leave no blank spaces to ensure a smooth reading. The results of section "2.1. Genomes quality assessment" should be better explained, as their importance in the development of the article is unclear.
- In section "2.2. Antibiotic resistance genes identification" I consider that it is not enough to perform a protein search in the Uniprot database to identify antibiotic resistance genes, therefore, I strongly recommend that PCR be performed with specific primers to ensure the presence of these genes in the samples analyzed. Based on this observation, I suggest that an image be created showing the most important results of the requested experiment.
- In section "2.3. Phylogenetic tree construction" Figure S1 is mentioned, but that image is not shown in the article and no supplementary document has been uploaded to the platform. Please add it to the article or upload it as a supplementary document. As I mentioned in the previous observation, searching for proteins in the Uniprot database is not comparable with searching for antimicrobial resistance genes using molecular biology methods, so what is mentioned in the paragraphs between lines 144 and 169 is not fully proven. Please perform the requested experiment to verify the veracity of what is mentioned in these paragraphs.
- In section "2.4. Antibiotic resistance assessment," I recommend improving the structure of Table 2, as it can be confusing for non-specialist readers.
- According to section "3. Discussion", because the methodologies used are not adequate, much of the information in this section lacks credibility, so I believe that the methodological design should be reconsidered and the information presented in this section should be subsequently considered.
- In section "4.1 Genomes and Bioinformatics Analysis," it is mentioned that DNA was extracted using the Magen HiPure bacterial DNA kit. I recommend providing a general description of the procedure performed, as well as the processing required for NextGenSeq, this will allow readers to understand the methodologies used in this research.
- According to section "4.2 Search for antibiotic resistance genes", I consider that this methodology is not adequate for searching for these genes in the strains analyzed. I suggest modifying the methodology used for one that allows for adequate verification of these genes.
- I believe that section "4.3 Bacterial strains and culture conditions" should be 4.1, because before explaining the bioinformatics analyses performed, the culture conditions of the strains used in the research must be fully understood.
- Section "4.5 Statistical Processing" should be renamed to "Statistical Analysis" and should be improved, considering that it should include a description of the data, the statistical analysis performed, and the processing used in the research to ensure that the "Materials and Methods" section is as complete as possible.
- The conclusion could be improved. I recommend briefly including the results of your research, its potential impact on animal and human health, and whether it adequately meets the research objectives, to further emphasize your results and their impact on public health.
Author Response
Dear respected Reviewer,
thank you for giving us the opportunity to submit a revised draft of our manuscript. We appreciate the time and effort that you and the reviewers dedicated to providing feedback on our manuscript and are grateful for the insightful comments and concerns on valuable improvements to our research article. We've made significant revisions to the manuscript and it's truly improved thanks to you!
The title of the article is very conflicting for me, as the search for antibiotic resistance genes is not being conducted through laboratory experiments, so I disagree with its appropriate title. I would also suggest "in Kocuria species," not "among Kocuria spp". In addition, it should be noted whether it was performed on ATCC strains, on meat samples, or on isolates from animals or humans.
We agreed. The name has been changed and the origin of the strains has been clarified.
I suggest rewriting the "Abstract" section to include background, methods, results, and conclusions. This will help readers properly understand what was done in the article.
We appreciate your feedback and have rewritten the abstract
Section "1. Introduction" should be improved. I recommend including more characteristics of Kocuria, the antimicrobial resistance mechanisms exhibited by this bacterial genus, and its impact on animal, human, and public health. The terms "flora" or "microflora" are obsolete; they should be replaced with "biota" or "microbiota". We have fixed the introduction and added several links to recent works. The terms fixed.
Section "2. Results" should follow section "1. Introduction" immediately; leave no blank spaces to ensure a smooth reading. The results of section "2.1. Genomes quality assessment" should be better explained, as their importance in the development of the article is unclear.
I apologize, but the publisher recently changed the order of the sections. We follow the publisher's template, so we can't make these corrections.
The results of section "2.1. Genomes quality assessment" should be better explained, as their importance in the development of the article is unclear.
This is a standard procedure of genome quality assessment and taxonomic placement verification.
In section "2.2. Antibiotic resistance genes identification" I consider that it is not enough to perform a protein search in the Uniprot database to identify antibiotic resistance genes, therefore, I strongly recommend that PCR be performed with specific primers to ensure the presence of these genes in the samples analyzed. Based on this observation, I suggest that an image be created showing the most important results of the requested experiment.
We consider that the presence of the sequences of these genes in the full genomes of our strains is a sufficient reason to conclude that these genes are present in analyzed samples. We extracted DNA from studied strains cultures and performed full genome sequencing using modern technologies. The quality and completeness of obtained genomes was high enough, so we can trust the data, obtained by analysing these genomes.
In section "2.3. Phylogenetic tree construction" Figure S1 is mentioned, but that image is not shown in the article and no supplementary document has been uploaded to the platform. Please add it to the article or upload it as a supplementary document.
Indeed, both supplementary documents did not load, we fixed it. All two supplementary files added.
As I mentioned in the previous observation, searching for proteins in the Uniprot database is not comparable with searching for antimicrobial resistance genes using molecular biology methods, so what is mentioned in the paragraphs between lines 144 and 169 is not fully proven. Please perform the requested experiment to verify the veracity of what is mentioned in these paragraphs.
We believe that full genome sequencing can be considered as molecular biology method and is sufficient to prove the presence of mentioned genes in our starins genomes.
In section "2.4. Antibiotic resistance assessment," I recommend improving the structure of Table 2, as it can be confusing for non-specialist readers.
Table 2 lists the names of antibiotics, briefly indicating their mechanism of action and the concentrations of these antibiotics that suppress the growth of bacteria for comparison with each other.
According to section "3. Discussion", because the methodologies used are not adequate, much of the information in this section lacks credibility, so I believe that the methodological design should be reconsidered and the information presented in this section should be subsequently considered
Thank you for such a careful attention to our work, but we disagree that the data we received lacks credibility. We consider that full genome sequencing followed by bioinformatic analysis is a reliable method to assesse the presence of special genes in the 5 studied samples.
In section "4.1 Genomes and Bioinformatics Analysis," it is mentioned that DNA was extracted using the Magen HiPure bacterial DNA kit. I recommend providing a general description of the procedure performed, as well as the processing required for NextGenSeq, this will allow readers to understand the methodologies used in this research.
We've updated the manuscript; a full description of the methodology can be found on the manufacturer's website. The work was carried out in full compliance with these instructions:
https://www.magen-tec.com/uploadfiles/2023%E8%AF%B4%E6%98%8E%E4%B9%A6DNA/D3146%20HiPure%20Bacterial%20DNA%20Kit.pdf
According to section "4.2 Search for antibiotic resistance genes", I consider that this methodology is not adequate for searching for these genes in the strains analyzed. I suggest modifying the methodology used for one that allows for adequate verification of these genes.
I believe that section "4.3 Bacterial strains and culture conditions" should be 4.1, because before explaining the bioinformatics analyses performed, the culture conditions of the strains used in the research must be fully understood.
Section "4.5 Statistical Processing" should be renamed to "Statistical Analysis" and should be improved, considering that it should include a description of the data, the statistical analysis performed, and the processing used in the research to ensure that the "Materials and Methods" section is as complete as possible.
The conclusion could be improved. I recommend briefly including the results of your research, its potential impact on animal and human health, and whether it adequately meets the research objectives, to further emphasize your results and their impact on public health.
We thank the reviewer for a very careful and meticulous reading of our work. We took into account numerous comments, which helped us improve it. We will not change methodology, as this will be a separate work. We will use some of the comments in the continuation of this work and in future articles. Thank you for your very detailed analysis of our work, it made the manuscript better!
Reviewer 2 Report
Comments and Suggestions for Authors
I have a few comments and recommendations on how the manuscript can be improved.
Introduction
- l. 42-48: it is not clear from this text whether the provided numbers refer to the cases of an individual experiment (and were these cases of infections or just bacterial isolates) or some larger study/studies?
- 87: aim of the study
The examinations were conducted on four and five strains of bacteria. Five strains is quite few to evaluate antibiotic resistance profile, as it may vary not only between the species, but also between the genera.
In general this study seems fine – it is methodologically sound and fits the aims of the journal, but it needs clearer emphasis on clinical relevance, more detailed discussion of unexplained resistance patterns, and – most importantly – clear explanation why the experiments have been conducted on the small number of strains.
Statistics - While MIC50 values are reported, there is limited description of experimental replicates or variability. A figure or table showing ranges or standard deviations would add transparency.
Several small English edits would improve readability (e.g. “else several isolates”
Some references could be updated to include the most recent AMR literature on Kocuria where available.
Comments on the Quality of English LanguageThe English is generally fine, there are some minor errors.
Author Response
Dear respected Reviewer,
we thank you for your thoughtful consideration and analysis of our work. Our responses to the comments are listed below.
1. «l. 42-48: it is not clear from this text whether the provided numbers refer to the cases of an individual experiment (and were these cases of infections or just bacterial isolates) or some larger study/studies?»
The lines have been changed to better understand the scope of the study.
2. «87: aim of the study»
This paragraph has been supplemented to draw the readers' attention to the different origins of the strains.
3. The examinations were conducted on four and five strains of bacteria. Five strains is quite few to evaluate antibiotic resistance profile, as it may vary not only between the species, but also between the genera.
Yes, we agree that five strains is not many. Unfortunately, obtaining additional strains proved impossible for a number of technical reasons. However, the advantage of our work is that four of the five strains have been virtually unstudied. Therefore, we hope that our study will be a valuable contribution to science (although I intend to expand our collection at the slightest opportunity to continue our research).
4. In general this study seems fine – it is methodologically sound and fits the aims of the journal, but it needs clearer emphasis on clinical relevance, more detailed discussion of unexplained resistance patterns, and – most importantly – clear explanation why the experiments have been conducted on the small number of strains.
Thank you for your opinion! We are currently preparing another article on biofilm formation by these bacteria (a very interesting and challenging subject), explaining this resistance. Our strains have different origins—two from a meat-processing plant and three from soil. We will emphasize this in the introduction and discussion to better justify our choose.
5. Statistics - While MIC50 values are reported, there is limited description of experimental replicates or variability. A figure or table showing ranges or standard deviations would add transparency.
Our bacteria form biofilms, which makes it difficult to assess antibiotic resistance using standard methods. We use this data representation because standard methods provide an inaccurate picture the data. A separate experimental paper on this issue is currently being prepared for publication.
6. Several small English edits would improve readability (e.g. “else several isolates”
We will use a language correction service to improve this deficiency.7. Some references could be updated to include the most recent AMR literature on Kocuria where available.
We agree that the body of work in this area is growing rapidly—we've added several new papers to the discussion. Thank you!
Round 2
Reviewer 1 Report
Comments and Suggestions for Authors
Dear authors, I consider it to be an interesting article, but some of the observations I previously shared have not been corrected, so I consider that the quality of the article is not yet sufficient for publication in the journal. Below I share several observations that will improve the quality of the article:
- The results of section "2.1. Genome quality assessment" should be better explained, as their importance in the development of the article is not clear.
- In section "2.2. Identification of antibiotic resistance genes" the results need to be better explained because it is not adequate. I also suggest creating a figure that outlines these important results. This will allow readers to properly understand the results obtained in the research.
- In section "2.4. Evaluation of antibiotic resistance", I recommend improving the structure of Table 2, since it seems broken, I suggest that it be reworked by improving the organization and format, this will allow readers to analyze it properly.
- In section "3. Discussion", the importance of the presence of antimicrobial resistance genes in Kocuria should be discussed in more detail, mentioning the impact this has on pathogenicity and public health.
- In section "4. Materials and Methods" a space should be added between each subtopic, this will allow readers to properly understand the information mentioned in this section.
- In section "4.1 Genomes and Bioinformatics Analysis", there is no brief explanation of DNA extraction using the Magen HiPure Bacterial Kit or an overview of the required processing, so I consider the section to be incomplete.
- Section "4.2 Search for Antibiotic Resistance Genes" should be explained in more detail, this will allow readers to properly understand the bioinformatics techniques used in the research.
- I consider that section "4.3 Bacterial strains and culture conditions" should be 4.1, since before explaining the bioinformatics analyses performed, it is necessary to fully understand the culture conditions of the strains used in the research.
- Section "4.5 Statistical Analysis" should be improved; a description of the data, the statistical analysis performed, and the processing used in the research should be included to ensure that the "Materials and Methods" section is as complete as possible.
Author Response
We thank the respected reviewer for his meticulous analysis of the manuscript. As a result of your meticulous work, the manuscript has been significantly revised and made much more understandable to readers. We have made corrections to the manuscript (uploaded to the portal), and we are attaching our responses to the list below.
1: [The results of section "2.1. Genome quality assessment" should be better explained, as their importance in the development of the article is not clear]
1: [Genome quality assessment. After genome assembly, the quality of the work must be assessed. This is a standard procedure that must be completed before beginning bioinformatic analysis to identify genes of interest. Genome assembly is assessed using metrics such as genome completeness (a measure of how much of an organism's complete set of DNA has been accurately reconstructed in a genome assembly), and contamination (is the presence of foreign sequences from another organism within a genome), which indicates the presence of genes in the assembly that do not damage the target organism. To assess the genetic relatedness between two organisms, digital DNA-DNA hybridization (dDDH) values ​​are calculated. This is a bioinformatics method used in microbial taxonomy to assess the genetic relatedness between two organismes, calculates the value based on a comparison of their genomes.]
2: [in section "2.2. Identification of antibiotic resistance genes" the results need to be better explained because it is not adequate. I also suggest creating a figure that outlines these important results. This will allow readers to properly understand the results obtained in the research]
2: [Thank you for your suggestion, to better understand this information, we have created a drawing and included it in the section]
3: [In section "2.4. Evaluation of antibiotic resistance", I recommend improving the structure of Table 2, since it seems broken, I suggest that it be reworked by improving the organization and format, this will allow readers to analyze it properly]
3: [In section "2.4. Evaluation of antibiotic resistance", I recommend improving the structure of Table 2, since it seems broken, I suggest that it be reworked by improving the organization and format, this will allow readers to analyze it properly.]
[Table 2 has been reformatted; I hope it will be more visual in the journal format.]
4: In section "3. Discussion", the importance of the presence of antimicrobial resistance genes in Kocuria should be discussed in more detail, mentioning the impact this has on pathogenicity and public health]
4: [Section "3. Discussion" has been expanded]
5:[in section "4. Materials and Methods" a space should be added between each subtopic, this will allow readers to properly understand the information mentioned in this section]
5:[Fixed]
6:[in section "4.1 Genomes and Bioinformatics Analysis", there is no brief explanation of DNA extraction using the Magen HiPure Bacterial Kit or an overview of the required processing, so I consider the section to be incomplete]
6:[Added]
7:[section "4.2 Search for Antibiotic Resistance Genes" should be explained in more detail, this will allow readers to properly understand the bioinformatics techniques used in the research]
7:[Added]
8:[I consider that section "4.3 Bacterial strains and culture conditions" should be 4.1, since before explaining the bioinformatics analyses performed, it is necessary to fully understand the culture conditions of the strains used in the research]
8:[Section 4.3 has been moved and renamed as 4.1]
9:[section "4.5 Statistical Analysis" should be improved; a description of the data, the statistical analysis performed, and the processing used in the research should be included to ensure that the "Materials and Methods" section is as complete as possible.]
9:[The nonparametric Mann-Whitney U-test was used to assess the significance of differences (they were considered significant at p < 0.05). The Mann-Whitney U-test, is used to assess the difference between two independent samples of continuous data. It compares the distributions of two independent groups to check whether one group tends to have higher or lower values than the other. It works by ranking all observations from both groups together and then evaluating whether these ranks differ significantly. It is used in cases when the data is not normally distributed and when the samples sizes are small.]
Reviewer 2 Report
Comments and Suggestions for Authors
All remarks have been referred to and corrections have been made where possible.
Author Response
We thank the highly respected reviewer for analyze the manuscript.
Round 3
Reviewer 1 Report
Comments and Suggestions for Authors
Dear authors, I believe this version of the article is of sufficient quality for publication in the journal. Congratulations. Best regards.